# Regression Analysis for COVID-19 Infections and Deaths Based on Food Access and Health Issues

**DOI:** 10.3390/healthcare10020324

**Published:** 2022-02-08

**Authors:** Abrar Almalki, Balakrishna Gokaraju, Yaa Acquaah, Anish Turlapaty

**Affiliations:** 1Computational Science and Engineering, North Carolina A&T University, Greensboro, NC 27411, USA; bgokaraju@ncat.edu (B.G.); ytacquaah@aggies.ncat.edu (Y.A.); 2Department of Electronics and Communication Engineering, Indian Institute of Information Technology, Sri City 517 646, India; anish.turlapaty@iiits.in

**Keywords:** COVID-19, GIS, machine learning, regression, North Carolina, Gilford County

## Abstract

COVID-19, or SARS-CoV-2, is considered as one of the greatest pandemics in our modern time. It affected people’s health, education, employment, the economy, tourism, and transportation systems. It will take a long time to recover from these effects and return people’s lives back to normal. The main objective of this study is to investigate the various factors in health and food access, and their spatial correlation and statistical association with COVID-19 spread. The minor aim is to explore regression models on examining COVID-19 spread with these variables. To address these objectives, we are studying the interrelation of various socio-economic factors that would help all humans to better prepare for the next pandemic. One of these critical factors is food access and food distribution as it could be high-risk population density places that are spreading the virus infections. More variables, such as income and people density, would influence the pandemic spread. In this study, we produced the spatial extent of COVID-19 cases with food outlets by using the spatial analysis method of geographic information systems. The methodology consisted of clustering techniques and overlaying the spatial extent mapping of the clusters of food outlets and the infected cases. Post-mapping, we analyzed these clusters’ proximity for any spatial variability, correlations between them, and their causal relationships. The quantitative analyses of the health issues and food access areas against COVID-19 infections and deaths were performed using machine learning regression techniques to understand the multi-variate factors. The results indicate a correlation between the dependent variables and independent variables with a Pearson correlation R^2^-score = 0.44% for COVID-19 cases and R^2^ = 60% for COVID-19 deaths. The regression model with an R^2^-score of 0.60 would be useful to show the goodness of fit for COVID-19 deaths and the health issues and food access factors.

## 1. Introduction

An outbreak is announced as a pandemic when it spreads in a large geographical area, infects, and results in mortality for a high number of people, and all of that is caused by a virus that is a subtype of a current virus [1]. The first pandemic recorded was in 1580 [1]. Before 1889, pandemics’ patterns show a 50–60-year cycle, while, after 1889, a 10–40-year cycle is shown, with the possibility of shortening [1]. Unfortunately, nothing has been done to change this pandemic pattern in the last century [1].

Research indicates that the current outbreak started to spread between people in late November to December 2019 [2]. On 31 December, 27 cases were recorded of unknown diseases [2]. The recent outbreak was identified on 7 January 2020, a virus called SARS-CoV-2, which is caused by the beta coronavirus and attaches to the lower respiratory census tract [2]. On 18 January, the cases spread around the country regarding the travel for the Chinese Lunar New Year [3]. The government started to lock down the city of Wuhan, considered as ground zero, and closed all routes to the province [3]. The origin of the cases was connected to visiting the Wuhan’s Huanan Seafood market [2]. All the cases were related to traveling from Wuhan until 2 February 2020 [3]. Later, the cases spreaded all over the world and to the United States of America. The first case in the United States was recorded on 20 January 2020 [4]. By October 2021, the United States recorded 44,518,018 total cases and 716,370 total deaths [4].

The Chinese government reacted to the spread of COVID-19 by restricting people’s movement, mandatory masks, and monitoring machines [5]. Internationally, the responses included things such as social distancing, vaccines, and disinfecting hands to control the spread [6]. The Center of Disease Control CDC in the United States reacted to the pandemic by advising mask use, requiring negative tests for people to enter the US from a foreign country, and collecting contact information from passengers to minimize incoming infection cases [7]. However, the World Health Organization recommendations of face masks and sanitizer were difficult to enforce in low-income countries in Africa because of poor facilities and low access to equipment [8]. 

Investigating the factors or variables associated with a pandemic is essential to understand its spread. In the case of this pandemic, investigating the COVID-19 spread in relation to food access distribution, income, population density, health issues, and poverty is associated with future pandemic recovery plans and prevention. Food access would be limited by a stay-at-home order, curfew, and social distance rule. At the same time, population density or human traffic in public places, such as food outlets, would increase the chance of infection. Food access in urban areas is a critical factor for human survival. Equal distribution of food outlets supports healthy and active life in communities, while unequal distribution may have a negative impact on people’s health and result in a higher incidence of diabetes and other health risks. Analyzing food distribution is a multi-variate problem as it depends on various factors of influence ranging from income to demography [9]. More variables, such as income levels, affect people’s ability to buy food and access transportation for takeout. Health issues and chronic diseases may be affected by the pandemic conditions and the consequences associated with weakened immunity and infections.

A healthy life and well-being are some of the United Nations goals and strategies, especially the Sustainable Development Goals SDG 3.3, which aims to end pandemics by 2030 [3]. However, the spread of a new virus threatens this goal [3] because this pandemic is not the first and will not be the last, and the frequency of these pandemics might increase as influenza mutates every cold season to form a new strain. Investigating the current stage of the pandemic and its adverse effects helps us as humans to prepare for future pandemics.

## 2. Literature Review

Scientists have documented outbreaks and pandemics and analyzed them to limit their negative influence. Previous pandemics, such as malaria and H1N1, affected human health and life. In a study by Malik & Abdalla, they mapped the spread of H1N1 by using spatiotemporal analysis. The study analyzed the spatial spread and spatial–temporal distribution with the factors of population density and international flights from Mexico [10]. The second study indicates the use of spatiotemporal analysis to map the H1N1 outbreak [11]. The study found that the virus infections did not spread much as clusters between the first and third weeks but increased to larger clusters in the sixth week [12]. These clusters started to converge further from week six to eighteen, and then started to decline in week 22 [12]. There have been some studies on pre-existing health risks and their susceptibility to higher infection rates during epidemics. One study presented the effect of obesity on influenza infection duration and concluded that obesity extended the shedding duration by 42% for influenza and by 43% for influenza-H1N1 [13].

Since COVID-19 was announced as a pandemic on March 11 2020, scientists started to study and analyze the spread of the virus and its associated factors. Several studies focused on the global scale, while other studies investigated smaller scales and examined specific variables’ correlation to COVID-19 [14]. In a study, the authors presented the sectors that were disrupted globally, namely: tourism, restaurants, leisure, entertainment, travel, sports, etc. [15]. Another study presented the comparison between developed and developing countries, where increased COVID-19 cases and deaths were present in developed countries compared to developing countries [14].

Pandemic spread and prediction could be analyzed by several methods, including the Geographic Information System (GIS) and machine learning (ML). The GIS is an effective tool for visualizing the spread of cases with spatial reference maps, time, location, and other overlaying techniques. The role of GIS is clear in mapping cases, mapping case clusters, mapping the outbreak spread, and helping decision-makers act [16]. The geospatial analysis of GIS on COVID-19 was mostly on five topics, which are spatial–temporal analysis, health and social geography, environmental variables, data mining, and web-based mapping [17]. As an example, GIS can be used for dashboard tracing, which was applied for the first time at John Hopkins University [18]. Another use of GIS was applied by the World Health Organization to illustrate confirmed cases and deaths [9]. More examples are in the HealthMap by the Boston Children’s Hospital, USA [19]. A study proved the effectiveness of ML models on outbreak predictions by applying multi-layered perceptron MLP and Adaptive Network-based Fuzzy Inference System ANFIS [20].

Currently, GIS is a useful tool for mapping cases and deaths, spreading, and predicting the future spread for health authorities regarding taking necessary and precise action on future outbreaks. The use of GIS is critical during the pandemic and post-pandemic for policymakers to make decisions on developing surveillance tracking systems for controlling and preventing future pandemics [18]. South Korea shows the best example of creating a web-GIS tracking for its pandemic tracking system by tracing cases and highly infected sites [21]. The application of the GIS into the South Korean method provided a decision-making tool on updated tracking and predicted the needed procedures [22]. Given this orientation, another study investigated the outbreak spree by applying the five GIS model sizes in the United States [23]. It investigated the differences in using different size modeling from local to global and applied those methods on four variables, black female populations, income, household income, and percentage of nurse practitioners [23].

More tools on the analysis of COVID-19 cases and spreads included statistical regression models. These models have been used to investigate the fluctuation in cases and then connect that to variables. A study presented the investigation in Germany on the spike and decrease in COVID-19 cases in the first two months of the pandemic and found increases and decreases in cases, and these changes on carve may be by variables that need to be studied [24]. More investigation on the correlation of COVID-19 with other variables, such as health issues, is critical around the world. The importance of cholesterol and its relation to the virus entering human cells is illustrated in a study, and lower cholesterol helps clear the virus sooner and limit infections [25]. High blood pressure recorded a correlation with a reduction in lung function [26].

The correlation of COVID-19 with variables has been investigated by several regression models as an efficient method. More specifically, research on the correlation with health issues has been applied and presents a correlation to various health issues. A multivariable linear regression analysis on global data of COVID-19 cases and deaths recorded a high correlation of cases and deaths with high cholesterol and high body mass [14]. Moreover, the correlation is stronger in the younger population [14]. An analysis in the United Kingdom on people’s body mass and COVID-19 hospitalization by applying logistic regression demonstrated higher hospitalization for people with obesity [27]. Further, a study conducted by penalized logistic regression models proved that hypertension illustrates a correlation to COVID-19 cases and mortality [28]. Nevertheless, moderate blood pressure is considered a dramatic factor in patient survival and limiting organ damage [28]. COVID-19 affects people’s health, and that effect may be more severe on people with health conditions. A study presented a regression analysis on patients’ clearance after being affected with COVID-19 and concluded that more days are recorded for people suffering from high cholesterol and diabetes [29]. Additionally, a study presented a positive correlation between COVID-19 and population density in India by computational correlation coefficient models [30].

The analysis of the geographical spread of a virus provides a tool for decision-making and long-term management for outbreaks [21]. Mapping the data based on normality would show a visualization, followed by normalizing data, such as showing the percentage based on every 100,000 people [31]. A recent study on the environmental effects of COVID-19 spread took place in China to analyze the effects of temperature and humidity [3]. The results illustrated the relationship between infected cases and weather, where low humidity supported the suitability and spread of the virus [24]. Moreover, strong cases showed a temperature range of 10 °C to 20 °C [3]. Furthermore, a higher number of cases were shown in economically developed cities, such as Beijing, and lower cases in less developed cities, such as Lhasa, which could be due to air pollution, geographical location, or population density [3]. Moreover, a study in Malaysia discussed that tourism was affected badly by the outbreak and, in turn, affected the economy and financial development [32].

Additional variables, such as income, were investigated in various geographical locations. An investigation in Spain demonstrated the negative correlation of the mean income to COVID-19 cases spread, where more cases spread in lower mean income districts because of low access to health care, lack of awareness, and poverty rates [33]. More specifically, low median income districts had 2.5 times higher cases than higher than mean income districts in Spain [33]. More studies presented the correlation of income to COVID-19 cases and deaths and its influence on food security. For instance, researchers in Kenya analyzed surveys on COVID-19 influence and concluded that low-income households that depend on labor jobs are more vulnerable to food insecurity due to financial shock [34]. More specifically, during the pandemic, people in low-income neighborhoods spent more time at work than those in high-income neighborhoods due to labor shortages [35].

COVID-19 has a long-term influence on food security and impacted a population increase of 17 million Americans in 2020 compared to 2018 [36]. Despite the increase in food insecurity, the pandemic has had a dramatic influence on the increase in children classified as having food insecurity by 3% more in 2020 than in 2017 [37]. Hence, the U.S government increased the free food programs in nation-wide K–12 public schools.

In Brazilian data studies, the investigations found a positive correlation to different socio-economic variables, such as population density, and negative correlation to social isolation rates, which proves the importance of social distancing enforcement [38]. Another investigation was done in India by statistical analysis called Pearson’s correlation coefficient [39]. A positive correlation between people density and COVID-19 cases was presented in five states. A statistical analysis recorded a correlation of COVID-19 with the number of tests and population density [39]. More variables, such as public transportation, were investigated for the correlation to COVID-19 cases and deaths. A statistical analysis recorded a correlation of COVID-19 with the number of tests and population density [40]. Regarding another study, a positive correlation was presented between public transportation sites, such as airports and train stations, and COVID-19 cases, in which the people living less than 25 miles from transportation spots showed higher cases than people living more than 50 miles away [41]. This was further supported by another study on the spatial distribution of COVID-19 cases in China, describing the possibility of transportation influence on the spread between neighborhoods [42].

The demographic variables were also investigated in several studies. A study that took place in the United States analyzed the cases and death numbers of COVID-19 and concluded that African Americans have the highest rates because of their low income, low access to transportation, and the high rate of chronic diseases, such as diabetes and obesity [43]. Also, the study recorded the vulnerability of the Hispanic community on the age of to the pandemic because of their high uninsured status rate, high chronic diseases, language barrier, and their immigration status [10].

Researchers indicate that there is a lack of application of GIS on pandemic spreads and more application is needed [12]. There is a need for more GIS analysis on the outbreak with different variables. Further research is needed to investigate more variables, such as food access, in the United States [11]. The proposed study illustrates the investigation of the spatial distribution of COVID-19 cases and deaths in Guilford County and examines the possibility of correlation with specific variables in food access and health risks. This study investigates variables such as health issues, income, food outlets and access areas, population density, and poverty rates. This study is applying technology by exploring machine learning models’ efficiency to analyze the pandemic distribution.

The research questions in this study are:Is it possible that COVID-19 cases and deaths in geospatial distribution are associated with food outlets and restaurants distribution?Can other variables illustrate a geospatial correlation with COVID-19 cases and deaths?How can machine learning discover a higher quantitative statistical correlation of COVID-19 cases and deaths against various independent variables?Do the machine learning results concur with the GIS regression results?

Our contributions in this study are:Investigated the geospatial association of COVID-19 cases and deaths to food outlets distributionExamined the dependency of various socio-economic and health risk variables on COVID-19 cases and deathsApplied ML techniques to investigate the statistical association between COVID-19 cases and deaths to other variables

## 3. Study Area and Materials

This study took place in Guilford County (Figure 1) in the state of North Carolina, with an area of 645.70 square miles and a population of 537,174 [44]. The county population consisted of 35.4% black, 49.4% is white, 5.3% Asian, 8.4% Hispanic, and 1.5% other [44]. The county took steps to maintain people’s health and wellness. Mandatory face masks were officially announced starting from 5 PM on Jun 26, 2020 [45]. Guilford County issued a “stay at home” order for transportation on April 17, 2020 [46]. In June, the county announced 5 testing sites spread around the county [46]. The county has three zip code areas with a high cluster of cases, and they are 27,405, 27,407, and 27,406 [47]. By October 14, 2021, North Carolina recorded 1,436,699 total cases [4]. The datasets were obtained from the health department in Guilford County.

## 4. Methods and Results

This study adopted a spatial-based and machine learning regression method to analyze the correlation between COVID-19 cases, deaths, and independent variables. The spatial method was applied to analyze the correlation and to present it visually on maps with variation of correlation degree. ML regression model is a strong tool that could be used for different topics and purposes, and the cause and analysis is one of them. Moreover, applying several models to compare results is important to find the most suitable model for this study and document it. In this study, the authors used ArcGIS-ArcMap software version 10.3 for GIS analysis and Jupiter software to apply the regression analysis. The method (Figure 2) applied used GIS tools for spatial and Sci-Kit Learn software libraries for machine learning regression, respectively. The GIS regression methods applied four models: the scatterplot matrix graph, spatial autocorrelation (Moran’s I), ordinary least squares (OLS), and the geographically weighted regression. The ML regression method applied four models, and they are linear multioutput regression, K-nearest neighbors of multioutput regression, random forest of multioutput regression, and support vector regression. These models were applied to analyze the correlation between dependent (COVID-19 cases and deaths) and independent variables (med-income, poverty rate, population density, high blood pressure, high cholesterol, obesity, number of healthy food outlets, and number of healthy food outlets).

### 4.1. GIS Methods

These maps in Figure 3 and Figure 4 present the COVID-19 cases and deaths. In Figure 3, higher numbers of cases are presented in dark blue color. The lowest COVID-19 infections are in the downtown of Greensboro, where it has fewer residential homes than businesses, and the highest are located outside of Greensboro in Summerfield, Gibsonville, Sedalia, Burlington, and Pleasant Garden. In Figure 4, the highest numbers of deaths are ranging between 22 and 33 per each census tract, displayed in blue color, and the lowest numbers of deaths are given 0 to 3 per each census tract in yellow color. The COVID-19 deaths low numbers are reported in Greensboro and the high mortality reported out of the city. An observation from this distribution could be about people’s education and the mask enforcement in large stores or offices. After that, scatterplot matrix graph in Figure 5 presents the interaction between COVID-19 cases and independent variables. The graph illustrates some positive and negative correlations and no correlation. Positive correlations include obesity with poverty and high blood pressure. Negative correlation is presented between obesity and med-income variables. However, there is no apparent strong correlation observed between COVID-19 cases and other variables through this scatter matrix visualization.

The scatterplot matrix graph is also applied to COVID-19 deaths as a dependent variable. The graph (Figure 6) also presents no correlation between COVID-19 deaths and variables. Negative correlations are presented between med-income and poverty and obesity.

After that, we applied the spatial autocorrelation (Moran’s I) to find the cluster of cases and deaths on some census tracts. The spatial autocorrelation is applied by this equation:(1)$$I=\frac{n}{S_{0}}\frac{\sum_{i=1}^{n}\sum_{j=1}^{n}Wi.jZiZj}{\sum_{i=1}^{n}Z_{i}^{2}}$$

In Equation (1) $Z_{i}$ is the deviation of an attribute for feature i from its mean ($X_{i}-\bar{X}$). The Wi.j is the spatial weight between feature *I* and j, and n is equal to the total number of features. The $S_{0}\;$is the aggregate of all spatial weight. After applying the equation, results are presented in Figure 7 and Figure 8. Figure 7 illustrates that COVID-19 cases are significantly clustered in Guilford County, which means there is high dependency of output and independent input variables.

In Figure 8, the spatial autocorrelation concluded that the cluster of COVID-19 deaths is a result of random chance, which encourages the investigation further on different variables. The Moran’s summary of COVID-19 cases and deaths by the Moran’s I spatial autocorrelation is in Table 1 below.

Next, local Moran’s was applied based on this formula:(2)$${I_{i}}_{\;}=\frac{\chi_{i}-\bar{X}}{S_{i}^{2}}\sum_{j-1,\;j\neq i}^{n}w_{i.j\left(xj-\bar{X}\right)}$$

In Equation (2), n is the total number of features, and $\chi_{i}$ is the attribute for feature i. Moreover, $w_{i.j}$ is the spatial weight between feature i and j. The output of this equation is presented in Figure 9 and Figure 10. Figure 9, the local Moran’s on COVID-19 cases, presents tracts with high case numbers and its correlation with a high number and percentage of variables in the south of Greensboro and east of Guilford County. The pink patch represents high cases of COVID-19 with an increase in variables. The red patch represents high cases and low variables correlation. The blue patch illustrates tract with low cases number with low variables in Greensboro downtown. In Figure 10, the local Moran’s on COVID-19 deaths is presented with the correlation of variables in each tract. The red patch represents high mortality with low correlation with variables, and the pink patch represents high mortality number with high variables in the north of Greensboro.

Then, OLS was applied to examine dependent and independent variables. OLS is a linear regression to perform a prediction or detect relationship between dependent and independent variables. We examine COVID-19 cases as a dependent variable with all independent variables. This OLS model uses the equation below:Y = β_0_ + β_1_ X_1_ + β_2_ X_2_ + β_n_ X_n_ + Ɛ(3)
where Y is the dependent variables, β is coefficients, X is explanatory or independent variables, and Ɛ is random error. In Figure 11, red patches represent areas with higher COVID-19 cases than the model predicted, and the blue shaded census tracts illustrate areas with lower COVID-19 cases than the model expected. In this model, the multiple R square was 0.358946, and the adjusted R-square was 0.307662. The Akaike’s information criterion (AICc) was 1412.247528. The joint F-statistic was 0.000000, which was a significant result. The Jarque–Bera statistic [g] was 1.511785, which indicates that the independent variables have an influence on the dependent variable. The joint Wald statistic [e] was significant and computed as 0.000000. The Keonker (BP) statistics, which determine if the independent variables have a consistent relationship to the dependent variable, was 0.009854, also significant, but the relationship is not consistent.

In Figure 12, red patches represent areas with higher COVID-19 deaths than the model predicted, and the blue shaded illustrates areas with lower COVID-19 deaths than the model predicted. In this model, the multiple R square was 0.159614, and the adjusted R-square was 0.092383. The Akaike’s information criterion (AICc) was 685.908921. Joint F-statistic was 0.021994, which was a significant result. The joint Wald statistic [e] was 0.000000 as a significant result. The Keonker (BP) statistics determine if the independent variables have a consistent relationship to the dependent variable, and it was 0.388493, which was not significant. The Jarque–Bera statistic [g] was 0.000000, which is significant and means the model is biased and needs further investigation.

Based on the independent variables’ coefficient of the OLS, variables with higher coefficients than 7.5 will be applied in the GWR. These variables are high cholesterol, high blood pressure, and healthy food outlets. In Figure 13 and Figure 14 GWRs were applied on COVID-19 cases and deaths to visualize the correlation with independent variables by applying this equation:(4)$$y\;={ℬ}_{0}+{ℬ}_{1}x+ℰ$$

In this equation above, the coefficient *ℬ*_1_ illustrates the increase in y because of one -unit increase in *x*. This map shows less tract with high correlation and more with medium correlation. In Figure 13, the map presents the correlation between the dependent and independent variables. Red patches, which represent high correlation, are in east of Gilford County in the tracts 012803, 015300, and 017200. In Figure 14, the map presents the correlation of COVID-19 deaths with variables (high cholesterol, high blood pressure, and health food outlets) and presents correlation degrees in color shades. The highest correlation of COVID-19 deaths with the variables is presented on the tracts 015703, 012604, and 013700.

### 4.2. ML Regression Results and Discussion

This study adopted machine learning techniques to investigate the correlation by applying both linear and nonlinear regression models. Linear, multi-output linear, random forest, and K-nearest neighborhood regression models were applied to investigate the data. All models investigate all variables at the same time, but linear regression investigates single output at a time. These four models were applied to evaluate their results. These models are predicting the values of the dependent variables, such as COVID-19 cases and COVID-19 deaths, with the correlation of independent variables of med-income, poverty rate, population density, number of healthy food outlets, and number of un-healthy food outlets. The dataset was divided into 80% training and 20% testing for multioutput model development. The training set contained eighty-seven (87) observations and twenty-two (22) observations in the testing set, and two different metrics: root mean square (RMS) and R-squared (R^2^), which were used to evaluate the models developed. The implementation of multioutput and multiple linear regression models were done with the Sklearn package in Python and MATLAB 2020a, respectively. The default parameters for the multioutput regression models were used in Table 2.

The equation below is derived in the linear regression model. In the equation, coefficients of variables were computed based on the linear regression model.
Y = 0.53 + 0.194 × 1 − 0.251X_2_ + 0.887X_3_ − 0.915X_4_ − 0.0996X_5_ + 0.315X_6_ − 0.026X_7_(5)

The degree of linear association between all variables is computed by the Pearson correlation coefficient (R^2^)-scores in the correlation matrix heatmap format in Figure 15. The results could be read in three directions: R values close to 1 show a positive relationship, and R values close to −1 illustrate negative relationships, but results close to zero have no linear relationships. It can be observed in the heatmap (Figure 13) that there is a positive correlation between obesity and poverty (R^2^ = 0.74). There is a high positive correlation between high cholesterol and high blood pressure (R^2^ = 0.82). Furthermore, there is a positive correlation between obesity and high blood pressure (R^2^ = 0.77). Moreover, there is a strong negative correlation between obesity and med-income (R^2^ = −0.7), and a negative correlation between income and poverty (R^2^ = −0.75). There is no correlation between COVID-19 cases and health issues (obesity, high cholesterol, and high blood pressure). Moreover, there is no correlation between unhealthy food outlets, healthy food outlets, and health issues.

From the tables’ results below (Table 3 and Table 4), the authors applied and compared the regression models results. The COVID-19 cases as a dependent variable have the highest value of R^2^-score as 45% by the application of linear regression for multioutput regression model, and COVID-19 deaths had a higher value of 60% by the application of support vector regression model. The high correlation R^2^-scores of COVID-19 deaths and variables were also presented by the GIS spatial autocorrelation as clustered distribution in Figure 7. These regression models’ results indicate that independent variables (med-income, poverty rate, population density, number of healthy food outlets, and number of unhealthy food outlets) have more influence on the dependent variable COVID-19 deaths than COVID cases.

The application of the multiple linear regression models considered the two dependent variables (COVID-19 cases and deaths). The support vector regression model was applied to examine all the data and errors within the threshold. In Figure 16, the predicted trends for dependent variable COVID-19 deaths are presented against the original trend values. Both trends, match the peaks and troughs well overall, showing similar behavior. However, the residual errors seem to vary both on the positive and negative side of the trend. The test data are kept out of the sample. The significance of Figure 11 is that the prediction trend is matching the peaks and troughs present in the original trend of number of COVID cases well (ground-truth). There are still many residual gaps between the original and predicted values, but the trend was predicted well overall. This figure coincides well with the R^2^-coefficient of 0.60 for number of COVID-19 deaths.

## 5. Conclusions

This study implemented GIS and machine learning techniques on COVID-19 data in 109 census tracts in Guilford County to investigate any correlation between the spread of the pandemic and social–economic, food access, and health issues variables. The GIS and machine learning methods were applied to examine the datasets and compare their results regarding if they are equivalent or different.

The GIS results illustrate the distribution of the variables where COVID-19 cases have a cluster in Guilford County, while COVID-19 deaths have no cluster. The cases cluster was biased and indicated more investigation of independent variables. COVID-19 deaths presented a *p*-value at 0.00000, which indicates a 99% confidence that independent dose had no influence on the distribution. Moreover, the COVID-19 infection cases *p*-value result was 0.516475, and that indicates less than 90% confidence that independent variables do not influence the distribution. The OLS results did not a indicate high influence of the independent variables on the dependents. The R-square of the influence of the independent variables on COVID-19 cases is only 35%. It also indicated only 9% on COVID-19 deaths. These percentages are low, and we suggest more investigation and including more variables.

The application of four spatial regression models indicates some influence on the independent variables. The heat map presented a weak correlation between the dependent and independent variables. There was a positive but not strong correlation between the dependent variables COVID-19 cases and deaths, which means deaths increase where cases are high. However, there were several strong negative correlations between income and two variables (poverty and obesity), but there was a positive correlation between poverty and obesity. More correlations between the independent variables are clear in a positive correlation of high blood pressure with obesity and high cholesterol. These independent variables do not show direct impacts on the dependent variables, but they affect people’s health, which could make them control variables. For example, poverty led to unhealthy diet, which affects people’s immune system, and the presence of two health issues in a community makes them more vulnerable to health issues and risks. The highest R-square for COVID-19 cases was 60% by support vector regression and for COVID-19 death; the highest R-square was 44% by the linear regression for multioutput regression. These numbers are not high for correlation, which indicates an unclear influence of the independent variables on the dependent variables.

The machine learning results take the same direction as the GIS results, correlation between variables or independent variables. The study illustrates the need for future investigation on the spread of COVID-19 infections and deaths in Guilford County. Further study may include the distribution of more health issues, such as autoimmune diseases, to investigate more correlations to COVID-19 infections. Further analysis would require more datasets or a larger geographical scale.

In future, this study would examine several variables exclusively independent in the regression model and investigate the feature engineering in machine learning to increase the R^2^-score. Other independent variables would be related to the distribution of health centers, religion, and public transportation stops and routes. These data could be obtained from the transportation department and state health department. This study has a data limitation. The study area has 118 census tracts but only 107 census tracts had all the data variables recorded. That affected the results because more data would show more correlation and distribution analysis. More data would provide a clearer picture of the analysis to examine the issues on a state level, which includes many counties, and to analyze patterns and compare the counties.

## Figures and Tables

**Figure 1 healthcare-10-00324-f001:**
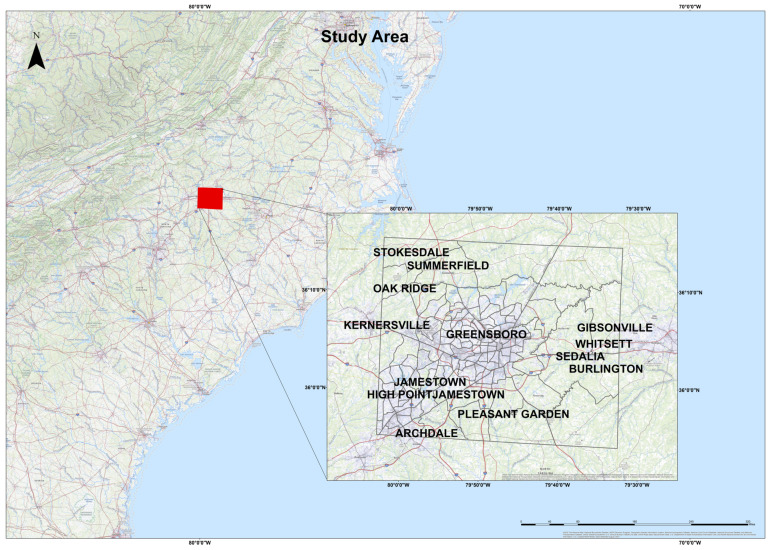
Study area.

**Figure 2 healthcare-10-00324-f002:**
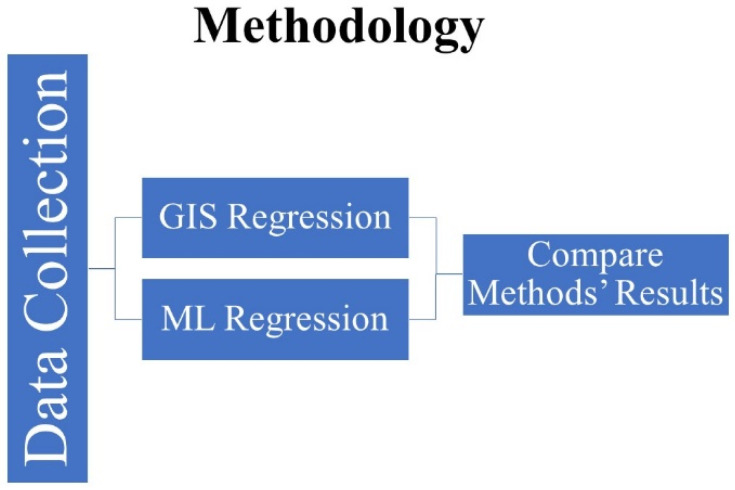
Methodology graph.

**Figure 3 healthcare-10-00324-f003:**
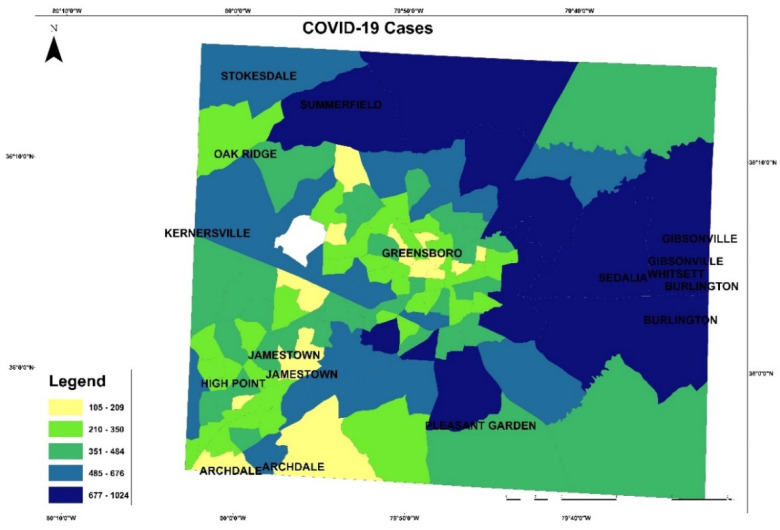
COVID-19 cases in Guilford County.

**Figure 4 healthcare-10-00324-f004:**
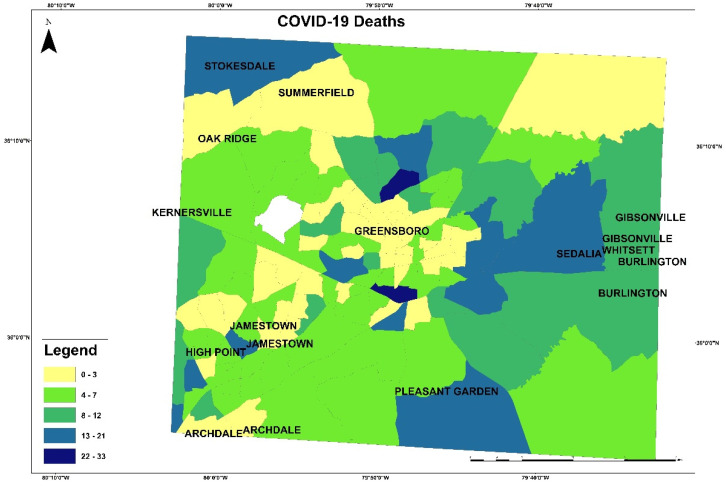
COVID-19 deaths distribution.

**Figure 5 healthcare-10-00324-f005:**
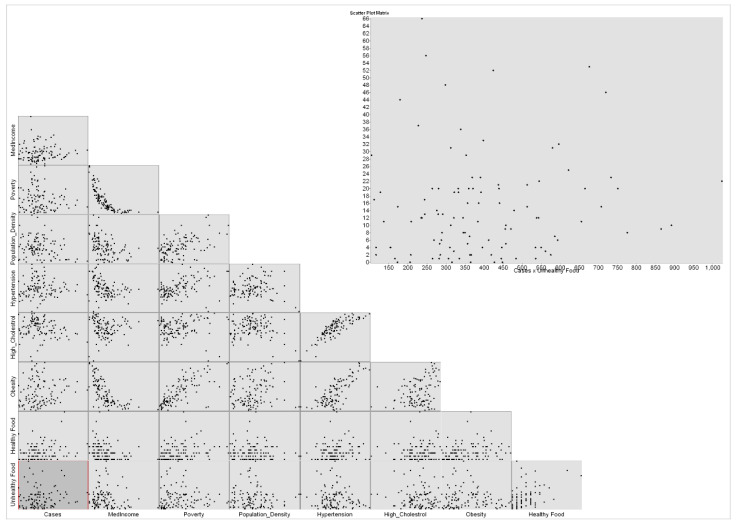
Scatterplot matrix graph using cases as dependent variable.

**Figure 6 healthcare-10-00324-f006:**
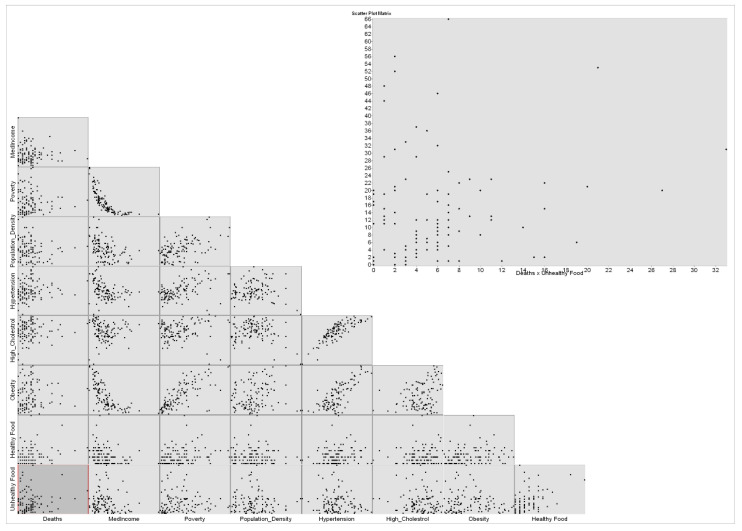
Scatterplot matrix graph using deaths as dependent variable.

**Figure 7 healthcare-10-00324-f007:**
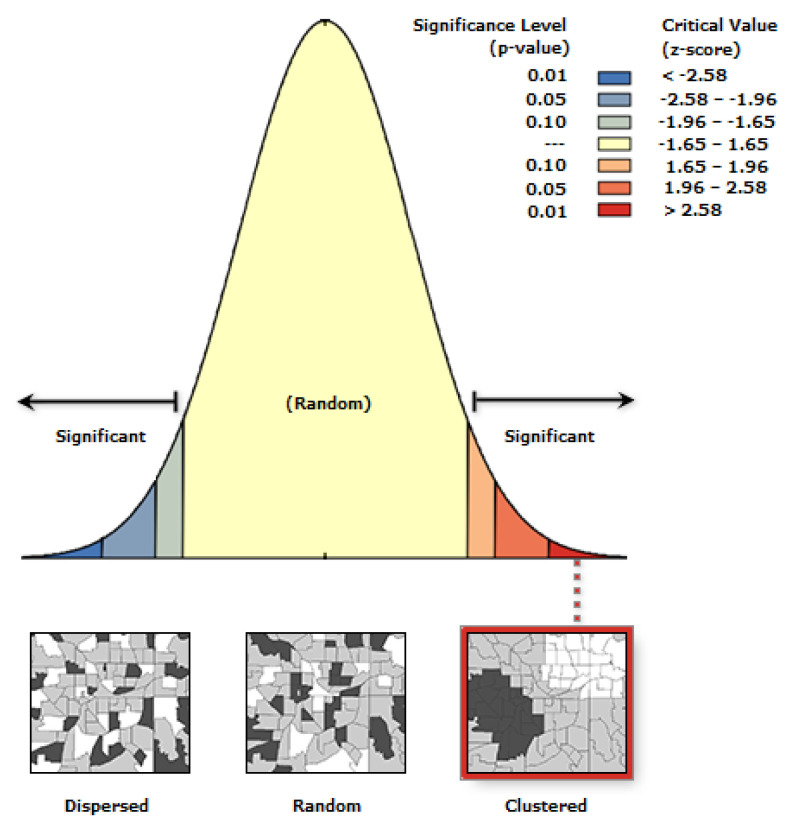
Spatial autocorrelation for COVID-19 cases.

**Figure 8 healthcare-10-00324-f008:**
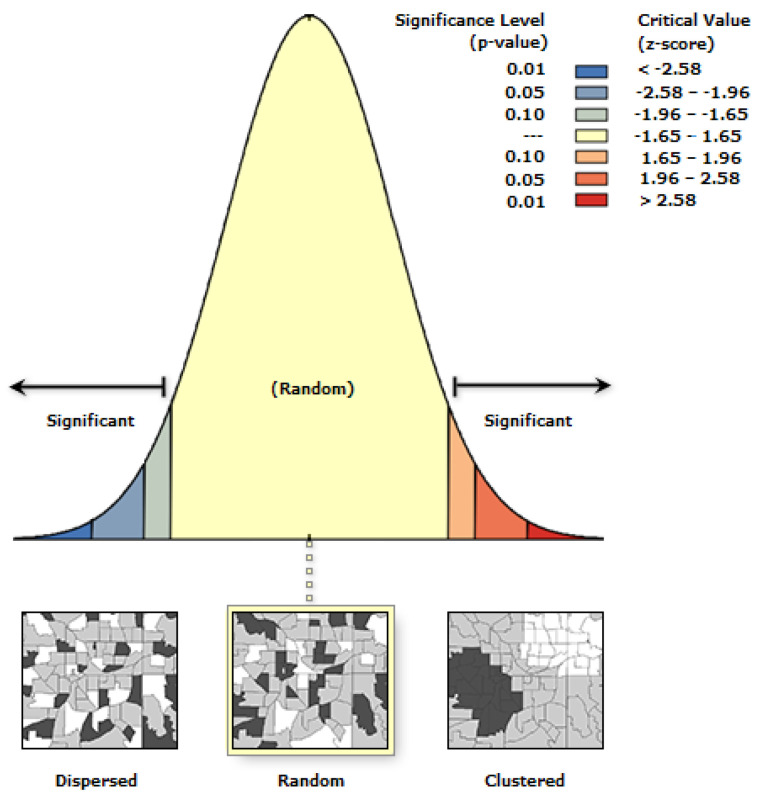
Spatial autocorrelation for COVID-19 deaths.

**Figure 9 healthcare-10-00324-f009:**
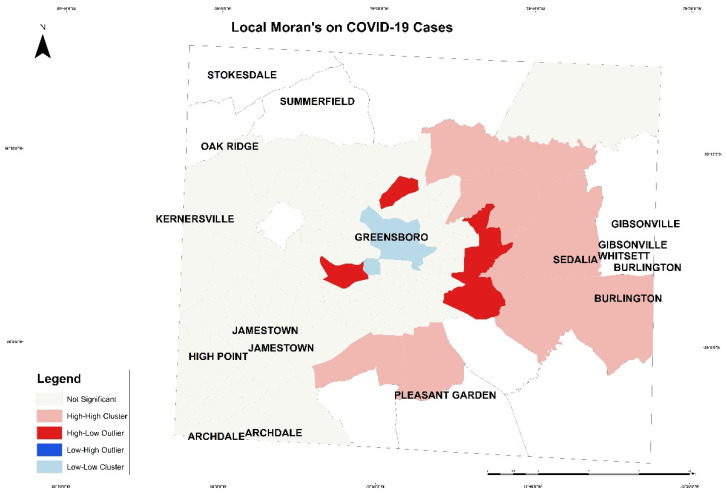
The local Moran’s on COVID-19 cases in Guilford County.

**Figure 10 healthcare-10-00324-f010:**
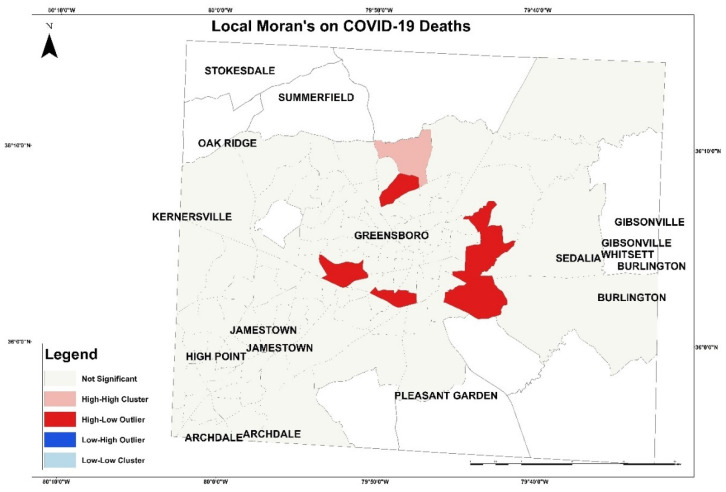
The local Moran’s on COVID-19 deaths in Guilford County.

**Figure 11 healthcare-10-00324-f011:**
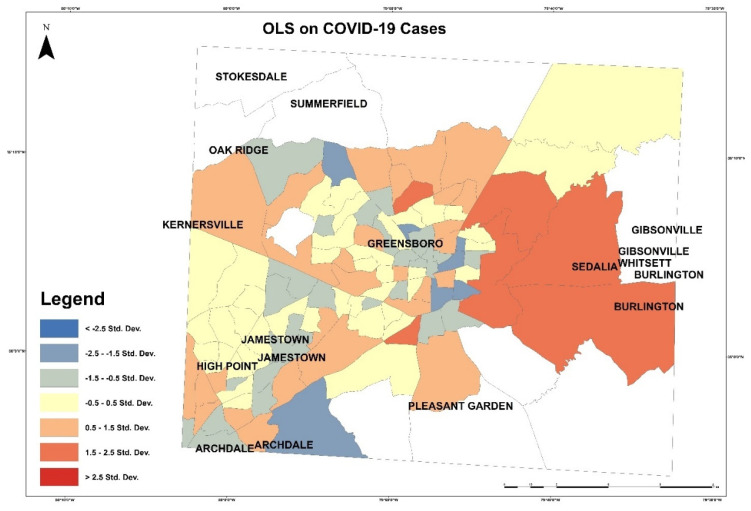
OLS on COVID-19 cases in Guilford County.

**Figure 12 healthcare-10-00324-f012:**
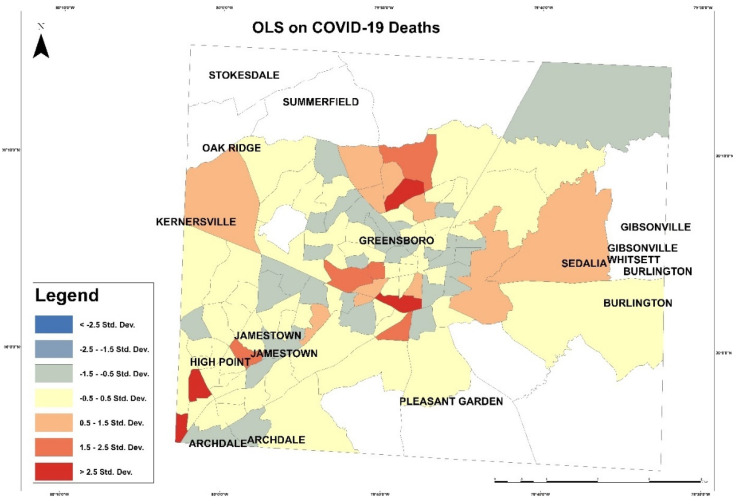
OLS on COVID-19 deaths in Guilford County.

**Figure 13 healthcare-10-00324-f013:**
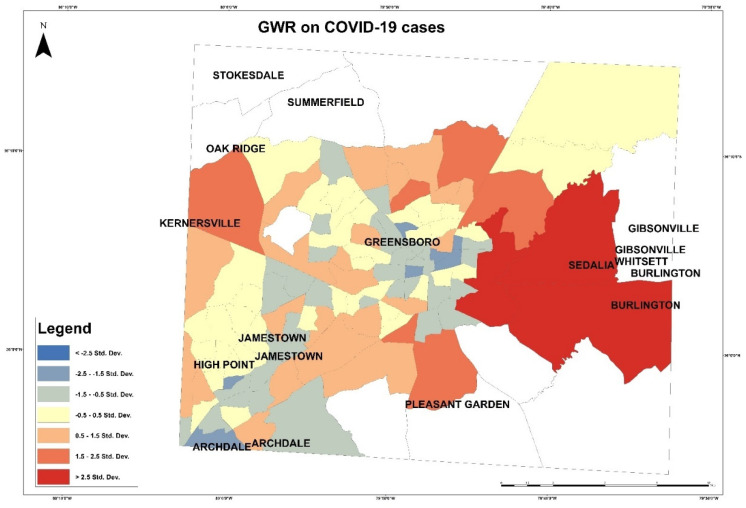
Geographically weighted regression on COVID-19 cases.

**Figure 14 healthcare-10-00324-f014:**
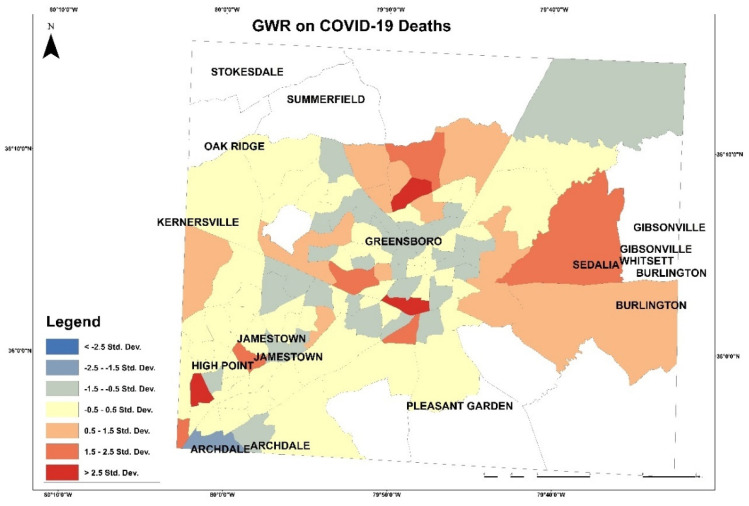
Geographically weighted regression on COVID-19 deaths.

**Figure 15 healthcare-10-00324-f015:**
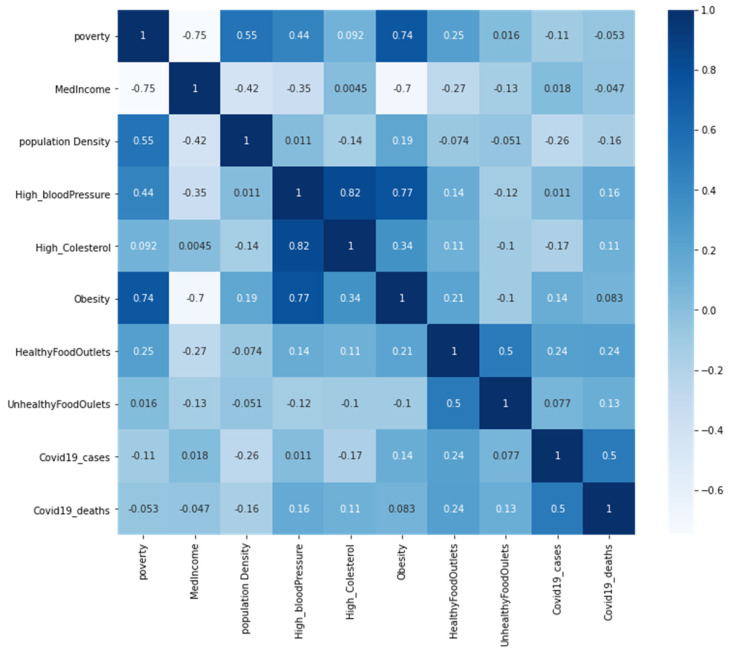
Correlation matrix with heatmap.

**Figure 16 healthcare-10-00324-f016:**
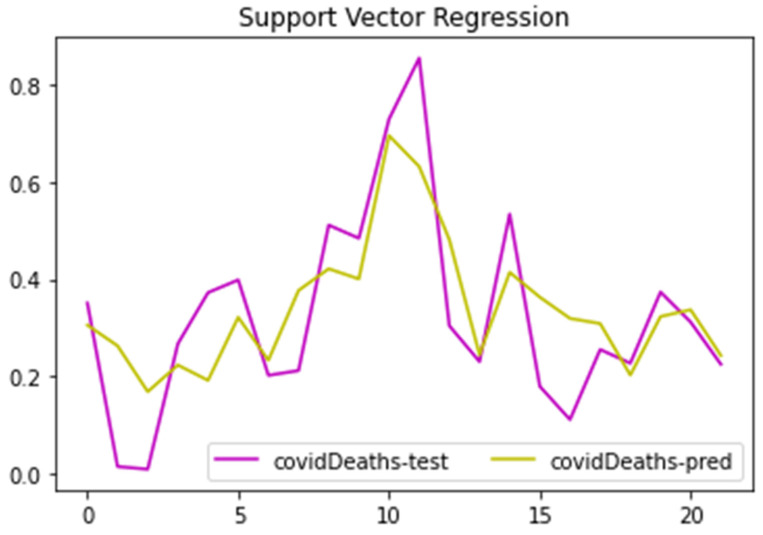
Support vector regression model.

**Table 1 healthcare-10-00324-t001:** OLS results for COVID-19 cases and deaths.

Measures	COVID-19 Cases	COVID-19 Deaths
Moran’s Index	0.118617	0.005965
Expected Index	−0.009259	−0.009259
Variance	0.000575	0.000551
Z-score	5.3314423	6.48788
*p*-value	0.000000	0.516475

**Table 2 healthcare-10-00324-t002:** Regression models’ parameters.

Model	Parameters
Linear Regression Model	copy_X = True,fit_intercept = True,n_jobs = None,normalize = False.
Random Forest Regression Model	bootstrap = True,ccp_alpha = 0.0,critrion = ‘mse’,max_depth = None,max_features = ‘ato’,max_leaf_nodes = None,max_saples = None,min_impurity_decrease = 0.0,min_imprity_split = None,min_samples_leaf = 1,min_samples_split = 2,min_weight_fraction_leaf = 0.0,n_estimtors = 100,n_jobs = None,oob_score = False,random_state = None,verbose = 0, warm_start = False)
K-Nearest Neighbor Regression Model	lgorithm’:’auto’,’leaf_size’:30,’metric’:’minkowski’,’metric_params’: None, ‘n_jobs’: None,’n_neighbors’: 5, ‘p’: 2, ‘weights’: ‘uniform’

**Table 3 healthcare-10-00324-t003:** R-square value of regression models.

Root Mean Square Error		
Models	CVID-19 Cases	COVID-19 Deaths
Linear regression for multioutput Regression	0.146	0.141
K-nearest neighbors for multioutput regression	0.208	0.147
Random forest for multioutput regression	0.186	0.175
Support Vector Regression	0.168	0.127

**Table 4 healthcare-10-00324-t004:** Root square error (RMSE) values of regression models.

Correlation Coefficient		
Models	CVID-19 Cases	COVID-19 Deaths
Linear regression for multioutput regression	0.446	0.508
K-nearest neighbors for multioutput regression	−0.085	0.466
Random forest for multioutput regression	0.137	0.239
Support Vector Regression	0.290	0.601

## Data Availability

Data such as income and food access were downloaded from the USDA Food Desert Locator Map website at https://www.ers.usda.gov/data-products/food-access-research-atlas/go-to-the-atlas.aspx (accessed on 28 October 2019).

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
