# Peer review of "Regression Analysis for COVID-19 Infections and Deaths Based on Food Access and Health Issues"

_healthcare, 2022, doi:10.3390/healthcare10020324_

Round 1

Reviewer 1 Report

The research is valuable, the following aspects are suggested to be modified and improved.

  1. Please supplement the main model formulas and descriptions used in the study.
  2. L270, ' Morans1 ' may be ' Moran's I '.
  3. L280,  ' Morn's ' may be ' Moran's '.
  4. P9, only the Global Moran's I is reported, the local Moran's I may be need to be reported.
  5. P11, The training set, verification set and prediction set of regression experimental data need to be explained in detail. In addition, the selection of optimal parameters also needs to be explained.
  6. Geographically weighted regression model may be more suitable for this study. Please supplement the experimental comparative analysis.
  7. Objectively speaking, are the machine learning methods used in this paper suitable for causal analysis? The robustness of the results is also questionable.
  8. In picture 11, what is the test data? and what is the pre data?  Are they in the sample or out of the sample? What is the research significance?
  9. The causal analysis of various methods is discussed further.
  10. The advanced experience of other countries or regions needs to be supplemented, such as China, especially the indicator of government control in causal analysis needs to be discussed.
  11. Please revise and supplement the references.

Author Response

The authors would like to sincerely appreciate the reviewer feedback and it noticeably improved the readability of the manuscript. We adopted all of your suggestions and listed the responses accordingly.

Reviewer 2 Report

The authors investigate the correlation between several socio-economic factors with SARS-CoV-2 spread in the Greensboro region of North Carolina. The authors present an initial literature review which I think fits well the proposed study. The authors have found no direct correlation between the socio-economic factors investigated and COVID-19 cases and deaths. However, the authors did find a relation between Greenboro areas and COVID-19 cases, but not for deaths. The research presented is straightforward and defines a possible method to investigate the outlined relationships. However, the authors need to correct/improve several aspects in the manuscript before acceptance.

1) I urge the authors to re-read the manuscript and correct for misspelt words and typos. I can safely state that there is almost one typo per line throughout the article, rendering some parts of the paper difficult to read. I found it entirely unacceptable that a manuscript is sent for publication in these conditions. It is a huge lack of care and dedication by the authors.

2) Figures should not be blurry or have unreadable or overlapped labels. The quality of the figures is poor. GIS figures are blurry and geographic coordinate labels are not readable. Labels in scatter plots should be corrected to human language instead of maintaining the typical column titles these datasets usually have. I invite the authors to pursue creative ways to enhance graphical representation instead of using the defaults from the plotting library used (some libraries have nice defaults though). Figure 7 and figure 8 are schematics of a concept, and, hence, they are repeated - they are not the result of a regression analysis. Figure 7 and figure 8 could be a single figure.

3) Authors should provide the scripts used to execute the calculations and explain which software they used in more detail. Authors can deposit their scripts on GitHub, or related platforms, so that other can use and reproduce their research easily, or adapt their methods to other regions of the globe.

4) line 218: percentages of the population do not sum to 100%

5) line 219: "weaknesses", did the authors mean this word?

5) "four models like", were these models used or similar ones "like" these? And if so, which? I think the word "like" renders the sentence confusing.

6) place figures 3 & 4 in the same figure, side-by-side, or near each other, to facilitate comparison

7) Figure 7 says COVID-19-related deaths are significantly clustered in Guilford County. But Guilford County is the whole area of analysis. It is not clear the comparison against which other region and where are the deaths clustered. Looking at figure 4, a cluster of deaths is not clearly visible. On the other hand, Figure 8 says COVID-19 Cases result of random chances. Looking at figure 3, cases do appear more clustered in the residential area of the County. Can it be that Cases and Deaths are swapped in the figures and text? 

8) The authors could explain in non-expert words the meaning of the OLS model and why regions deviate from the prediction. It is unclear to me which data was left out to create the model and which data was used to understand that the model and the data deviate in certain regions.

9) Conclusions in lines 329 to 331 are good controls to validate the method used because correlations between poverty and obesity, cholesterol, and high blood pressure are well known. The authors could emphasize this as a control to add value to the method.

10) The authors could extend the explanation in figure 11. However, it is now clear, nor explicit, which conclusions and consequences to the paper come from the fact that "the residual errors seems to vary both on positive and negative side of the trend".

11) line 372: authors suggest including more variables in additional studies. Which other variables could be included? Where could data for these variables be found, and why did the authors not use such variables for this study if they have a pipelined method?

12) line 377: the authors indicate a positive yet not strong correlation between the independent variables (COVID-19 cases and deaths). Why is the correlation not strong? And, which variables could the authors include to investigate this lack of absolute relationship? Is it access to better health care? Proximity to hospitals? Religion?

13) To understand the dependency of the correlation with the data availability, authors could perform a repetition essay leaving parts of the data out to understand the variability of the results on the lack of data. This "leave-data-out" essay would support the last standard claims saying the more data, the better. The authors had available 107 census tracts with complete data, not a small number.

14) Overall, I suggest adding additional explanations on the rationale of the method. Because this is a paper related to COVID-19, it is urgent to make it understandable to the broadest public as possible.

Authors should address these points before the paper is considered for publication.

Author Response

(The authors gave the same response as above.)

Round 2

Reviewer 1 Report

The research is of great scientific and practical significance. The author has made sufficient modifications. It is recommended to be publicated.

Reviewer 2 Report

The authors have addressed my concerns. The manuscript now reads much clearer, and the additional figures are appreciated. Also, the methods are explained in more detail. I have just identified some minor typos that can be easily corrected in lines:

line 57
line 243
line 286
line 357
line 364
line 392
line 394
line 417

After that, I propose the article for publication.